# Gait Recovery in Spinal Cord Injury: A Systematic Review with Metanalysis Involving New Rehabilitative Technologies

**DOI:** 10.3390/brainsci13050703

**Published:** 2023-04-22

**Authors:** Giuseppe La Rosa, Marianna Avola, Tiziana Di Gregorio, Rocco Salvatore Calabrò, Maria Pia Onesta

**Affiliations:** 1Consorzio Siciliano di Riabilitazione, 95100 Catania, Italy; larosa.csr@outlook.com (G.L.R.); mariannaavola.md@gmail.com (M.A.); 2Unità Spinale Unipolare, AO Cannizzaro, 98102 Catania, Italy; digregoriotiziana.doc@gmail.com (T.D.G.);; 3IRCCS Centro Neurolesi Bonino-Pulejo, 98124 Messina, Italy

**Keywords:** gait recovery, spinal cord injury, robotic rehabilitation, intermittent hypoxia, transcranial magnetic stimulation, transcranial direct current stimulation

## Abstract

Gait recovery is a fundamental goal in patients with spinal cord injury to attain greater autonomy and quality of life. Robotics is becoming a valid tool in improving motor, balance, and gait function in this patient population. Moreover, other innovative approaches are leading to promising results. The aim of this study was to investigate new rehabilitative methods for gait recovery in people who have suffered spinal cord injuries. A systematic review of the last 10 years of the literature was performed in three databases (PubMed, PEDro, andCochrane). We followed this PICO of the review: P: adults with non-progressive spinal cord injury; I: new rehabilitative methods; C: new methods vs. conventional methods; and O: improvement of gait parameters. When feasible, a comparison through ES forest plots was performed. A total of 18 RCTs of the 599 results obtained were included. The studies investigated robotic rehabilitation (*n* = 10), intermittent hypoxia (N = 3) and external stimulation (N = 5). Six studies of the first group (robotic rehabilitation) were compared using a forest plot for 10MWT, LEMS, WISCI-II, and SCIM-3. The other clinical trials were analyzed through a narrative review of the results. We found weak evidence for the claim that robotic devices lead to better outcomes in gait independence compared to conventional rehabilitation methods. External stimulation and intermittent hypoxia seem to improve gait parameters associated with other rehabilitation methods. Research investigating the role of innovative technologies in improving gait and balance is needed since walking ability is a fundamental issue in patients with SCI.

## 1. Introduction

According to the WHO (World Health Organization), spinal cord injury (SCI) is defined as any “damage to the spinal cord resulting from trauma or disease or degeneration” [1], representing one of the most devastating and debilitating conditions that an individual can sustain [2]. It interrupts the physiologic conduction of the nervous signal through the descending (motor) and ascending (sensitive) paths between the brain and periphery. Depending on the entity and localization, SCI may result in two main clinical conditions, i.e., paraplegia or tetraplegia, and both may be complete or incomplete with several combinations of symptoms affecting patient quality of life [3]. Sphincteric [4] and autonomic [5] involvement may occur as well.

One of the main goals of patients with SCI is to restore ambulation [6] since reduced mobility may affect psychological well-being [7]. Walking, as stated by Jaquelin Perry, is “is the body’s natural means of moving from one location to another. It also is the most convenient means of traveling short distances.” [8] However, it is a “social means” fundamental to meeting other people and attending personal and social activities. Gait is an apparently simple motor function, i.e., a “voluntary” movement, regulated by an automatic process, which is evoked by sequential activations of neurons in the brainstem and spinal cord [9]. Gait control requires the activation of the entire nervous system and musculoskeletal system [10]. In the last few years, advances in neurobiology and in the neurorehabilitation field have led to a better understanding of the effects of SCI on gait function and, therefore, in developing treatments to ameliorate the outcomes [11]. Lately, the most relevant mechanism of walking recovery in humans with SCI was related to the concept of central pattern generators (CPGs) [12]. Following this theory, gait can be recovered through rhythmic stimulation, which is a substantial part of different rehabilitation methods [13].

To date, gait rehabilitation protocols for SCI treatment include physical therapy [14] with non-invasive interventions, consisting of various types of motor training that are proven to decrease the inflammatory response, increase neurotrophin levels, and may strengthen spared functions and guide spinal rearrangement [11]. A conventional training program primarily provides compensatory strategies to regain motor activity and recover from damages of the spinal cord [15]. These strategies can be administered alone or combined with different technology-aided modalities such as treadmill training, overground training, body weight-supported gait training [16], robot-assisted gait training(RAGT) [17], and exoskeletons [18]. These methods seem to work, according to the CPGs theory, both with rhythmic ambulation stimulation (treadmill training, overground training, and body-weight-supported gait training) and reproduction of the rules of spinal locomotion (exoskeletons and robot-assisted gait training) [13]. Other strategies, such as external stimulation [19,20] and intermittent hypoxia [21], aim to enhance and stimulate neural plasticity and have been analyzed and combined with other rehabilitation technologies to improve mobility in SCI patients. 

The aim of this systematic review is to investigate the emerging technologies and strategies for gait recovery in adults with SCI, with particular interest in robot-assisted rehabilitation. 

## 2. Materials and Methods

This review follows the PRISMA [22,23] and PICO [24] criteria, as shown in Table 1.

To meet PICO criteria, a search was conducted on some of the major scientific databases (Pubmed, Cochrane, and PEDro), identifying randomized trials (RCTs), published in the last 12 years in English, and investigating innovative rehabilitation programs for walking recovery in people with SCI. Previous systematic reviews were excluded since we wanted to analyze the row data of the included study by ourselves. Only the first parts of crossover trials were included to avoid risk of overlapping results between the two methods. Association of more than one therapeutic device was accepted, if it drew comparisons with conventional rehabilitation. 

### 2.1. Literature Search Strategy

A literature search was performed on 29 July 2022 using Pubmed (https://pubmed.ncbi.nlm.nih.gov/) with the following search string: “(spinal cord injury) AND ((recovery) OR (outcome) OR (prognosis)) AND ((ambulation) OR (walking) OR (walk) OR (gait))”. Data from 2010 to 2022 from studies containing RCTs and clinical trials were considered, totaling 151 results. Another literature search was performed on August 8 2022, using the PEDro database (https://pedro.org.au/) with the terms “spinal cord injury” gait. Data from 2010 from papers containing trials only were considered, totaling 23 results. A third literature search was performed on the 10 August 2022 using the Cochrane website (https://www.cochranelibrary.com/) with the search string “((spinal cord injury) AND ((recovery) OR (outcome) OR (prognosis)) AND ((ambulation) OR (walking) OR (walk) OR (gait))):ti,ab,kw (Word variations have been searched)”. Publications dated between January 2010 and December 2022 that contained trials (word variations were searched)as well as included ICTRP 65 and CINAHL 39. totaled 117 results from Pubmed, 128 from Embase, and 103 from CT.gov. 

### 2.2. Inclusion/Exclusion Criteria

Inclusion criteria included RCTs and Crossover Trials published between 2010 and 2022, studies written in the English language, trials involving subjects older than 18 years of age, trials using the walking-outcome measure, trials comparing at least one innovative rehabilitation method versus a conventional method, and trials involving people with an SCI of any level and non-progressive nature. Studies investigating treatments involving a drug only, analyzing only a conventional rehabilitation method, involving a progressive spinal injury, or involving patients younger than 18 years old were excluded.

### 2.3. Data Extraction and Criteria Appraisal 

All data were extracted from article texts, tables, and figures. Three independent authors reviewed each article (G.L., M.A., and T.D.). Discrepancies between the three reviewers were resolved through discussion and consensus. The results of every stage of selection were reviewed by the senior investigators (R.S.C. and M.G.O.).

### 2.4. Risk of Bias

Risk of bias was evaluated by one of the authors (M.A.) through the RoB 2 [25] method, following the Cochrane Library [26] guidelines. Domains D1, D3, and D4 that involved adherence to intervention were investigated. 

### 2.5. Data Analysis and Heterogeneity of the Studies

Where necessary, data expressed as Standard Error (SE) were converted to Standard Deviation (SD), which was more manageable, through the formula SD=SE ∗√N, where *N* stands for the statistic sample. Studies showing results as median and interquartile range were converted to mean ± standard deviation (SD) according to the method of Wan et al. [27]. 

Effect size was analyzed, not strictly, because of the low heterogeneity of the outcomes evaluated (I^2^ = 0) and wide confidence intervals (CI 95%). Forest plots were generated, where possible, for more clearness regarding effect size, using the statistic software ProMeta 3 by (https://idostatistics.com/prometa3 3.0 ed. Internovi Cesena, Italy accessed on 23 January 2021). Effect sizes were acquired by comparing arithmetic means and SD pre and post intervention of the outcomes through the “Hedge’s G” method. The statistical model adopted was the “Random effect model”. The statistical analysis was performed by the first author (G.L.).

## 3. Results

A total of 599 articles were collected from the databases. The selection process is described in Figure 1, using the 2020 version of the PRISMA Flow diagram for new systematic reviews [28].

Eighteen RCTs were included in the systematic review. Study characteristics, data, and outcomes are presented in Table 2. Trials were sorted into three categories. The first category included studies involving the use of robotic devices such as exoskeletons on a suspension treadmill (n = 7) [29,30,31,32,33,34,35] (Lokomat). One of these studies evaluated the use of such devices associated with electromyography (EMG) feedback [36] (n = 1) (3DCaLT), and two evaluated overground exoskeletons [18,37] (n = 2) (Ekso). In the second group, we included studies assessing the use of intermittent hypoxia (IH) vs. normoxia (NX) during traditional gait training [38,39] or during suspension treadmill training [40] (BWSTT) (n = 3).The third group included studies that used magnetic [41,42,43] (n = 3) (TMS) or electrical [44,45] (n = 2) (tDCS) external stimulation during robotic or conventional rehabilitation. 

After a preliminary data analysis, authors agreed to investigate studies with different approaches due to a lack of data to perform a full metanalysis. Metanalysis for studies of the first group involved 10MWT [46] for gait speed, LEMS [47] to measure strength in key muscles of lower limbs, and WISCI-II [48] and SCIM-3 [49] to evaluate gait autonomy in the first group studies (Figure 2, Figure 3, Figure 4 and Figure 5, summarized in Table 3). Sequentially, a narrative review for the rest of data we collected is shown in Table 4.

Evaluations of all the studies before and after intervention are summarized in Table 3 and Table 5 as arithmetic mean ± standard deviation (SD). 

Regarding the studies of this group, we also decided to extract the dosage of robotic treatment in Table 5 to facilitate further comparisons among robotic and non-robotic methods. The risk of bias analysis in the studies is summarized in Figure 6 through “traffic light” graphics.

## 4. Discussion

This systematic review investigates the use of innovative rehabilitation approaches in patients with SCI. For the first time ever, we have gone beyond robotics since our study also focuses on other promising protocols, paving the way for more personalized training. Indeed, previous reviews have analyzed the effect of singular rehabilitation devices or type of technologies, focusing on stationary robotic-assisted gait training [16,50,51,52,53], and overground exoskeletons [54,55,56], leading to controversial results. 

A review by Cheung et al. highlighted improvement in independence and endurance linked to robot-assisted gait training in patients with SCI [53], while Alashram et al. found an increase in gait speed, walking distance, strength, range of motion, and mobility after incomplete SCI through the use of the Lokomat device [50]. Less encouraging results were found in more-remote reviews, such as the one by Swennen et al., in which no evidence on the improvement of walking function after robot-assisted gait training was found [16]. At that time, well-designed randomized controlled trials were lacking, and more evidence was needed. 

Regarding rehabilitation through the use of powered exoskeletons, different results were found in complete and incomplete SCI in a recent scoping review. Walking performance tested with 10MWT, 6MWT, TUG, and WISCI improved only in patients with incomplete SCI [56]. However, as demonstrated by Louie et al., only powered exoskeletons can provide the ability to walk at modest speeds to non-ambulatory individuals with complete SCI [55].

No other reviews explore the use of external stimulation and intermittent hypoxia in gait rehabilitation in people with spinal cord injuries, possibly because of the lack of comparable outcome data. This was why we were able to perform a metanalysis only on robotics and for specific outcomes (i.e., 10MWT [46], LEMS [47], WISCI-II [48], and SCIM-3 [49]).

### 4.1. Studies on Robotics 

Technological devices, such as robots, are able to deliver repetitive, high-intensity, standardized movement [57], a task that is difficult to achieve through manual therapy. Our analysis demonstrated that there is a sufficient magnitude [58] of WISCI-II with ES (Effect Size) = 0.23 (ES > 0.2) for robotic rehabilitation methods. No other relevant results in terms of ES were observed for other outcome measures. In our analysis, we did not observe a correlation between the exposition to the intervention and ES. Nevertheless, data showed that a major dosage of intervention leads to the highest ES, as shown in Table 4. The contribution of the other outcome measures is uncertain, as the ES of robotic rehabilitation is high but with an insufficient magnitude. 

Two studies evaluated the use of an exoskeleton (the Ekso^®^): one comparing it with conventional physical therapy [18] and the other comparing it with conventional therapy and BSWTT or standard care [37]. The first one was a pilot study on feasibility and efficacy, wherein all participants were able to complete the training, and significant improvement in the stride length, right step length, and 6MWT were observed after the experimental training. The second study confirmed that gait training with Ekso is safe and feasible in an outpatient setting, increasing gait speed (even though not statistically significantly), leading to clinically significant improvement proved by the transition in the gait speed category from home to community ambulation. Wearable exoskeletons, therefore, can be utilized as a gait-training device to facilitate motor and gait function recovery and health promotion for people with SPI. To this aim, a new prototype using pneumatic actuators may be used. Indeed, a geometrical model of a pneumatic exoskeleton based on anthropometrical parameters has been proposed. The authors attempted to demonstrate that their approach can be utilized to analyze the kinematics of the positions of lower limb segments and exoskeleton elements. Then, by changing every kinematic parameter (angle and angular velocity of each joint), how actuators should work can be observed [59].

### 4.2. External Stimulation (Non-Invasive Brain Stimulation)

The five studies selected evaluated two different methods of brain stimulation: Transcranial Magnetic Stimulation (TMS) and transcranial Direct Current Stimulation (tDCS). The TMS functioning mechanism seems to be activated by two different pathways, such as recruitment of neural networks [60,61,62], resulting in stimulation of neuroplasticity and modulation of the cerebral cortex with a reflection on descending paths [19,20]. These methods were associated with physical therapy, and efficacy in gait recovery was investigated. TMS associated with conventional physical therapy showed a 12.4% increase in speed in the group treated with TMS + physical therapy, a 16.5% increase in speed in the group treated with placebo + physical therapy, and an 18.7% increase in speed in the group using TMS only. Real TMS compared with SHAM was associated with a greater increase in total leg maximum voluntary contraction, with no significant results in gait parameters [43]. The association of TMS with gait training using Lokomat in sub-acute patients has proven more effective. Indeed, at follow-up evaluation, 71.4% of patients receiving TMS could complete the 6MWT, whereas only 40% of the sample treated with placebo + Lokomat could complete it. Nonetheless, no improvement in WISCI-II was observed in the two groups.

In contrast, robotic rehabilitation using Lokomat coupled with tDCS was applied in the study by Raithatha et al. [44], wherein the experimental group showed better results in evaluation in terms of strength measured by Manual Muscle Testing (MMT) with intragroup changes from baseline to postintervention of 4.3 ± 2.1 on the left lower limb and 9.1 ± 3.8 for the right lower limb. Previous research has shown that anodal tDCS applied to the motor cortex increases corticospinal excitability in both healthy adults and stroke survivors [63]. Moreover, it seems that neuromodulation may improve the acquisition and consolidation of motor abilities [64], though more recent research has shown a substantial degree of interindividual heterogeneity in motor cortical responsiveness to tDCS [65], which may contribute to a lack of effect observed in other studies. Taken together, data on these coupled approaches suggest that it is possible to associate rehabilitative methods (physical therapy and robotics) with tDCS and TMS to further improve functional recovery in patients affected by SCI.

### 4.3. Intermittent Hypoxia

Intermittent hypoxia (IH) appears to increase physical performance of people with incomplete SCI, as shown by Trumbower in al. [66]. This method seems able to stimulate synaptic plasticity of nervous fibers spared after injuries through the secretion of serotonin from Carotidis chemoceptors [67,68]. The release of serotonin activates the specific receptor 5-HT2, triggering the release of BDNF (brain-derived neurotrophic factor) [69] and the activation of the tirosin-kinasis signal path [70]. Some authors [39,40] tried to confirm the hypothesis of a better outcome by associating a rehabilitation treatment to IH. 

Navarrete-Opazo et al. [40] associated BWSTT with a 90” IH FiO_2_ 9%, 15 times a day for 5 days in a row, followed by IH 3 times a week for the following 3 weeks. Gait training sessions were executed 3 times a week for 3 weeks. Beginning on the fifth day of IH therapy, an increase in speed measured through the 10MWT was observed; moreover, this gain was maintained for the next 3 weeks, and endurance increased as well. 

Another study by Hayes et al. [38], applied the same protocol for 5 days in a row and was associated with 30 min of walking on the ground one hour later. The most important result was an endurance and speed increase (as per 6MWT and 10WMT, respectively) in the group treated with IH + walking.

Finally, it has been shown that IH elicits clinically meaningful improvements in walking speed and distance that persist for weeks after treatment [39]. However, given the lack of common outcome measures and the few patients enrolled in these published studies, it is not possible to state whether and to what extent IH works for treating SCI.

## 5. Conclusions

A cohort of new rehabilitation methods have been studied in the last 12 years to improve gait recovery in SCI patients. These can be associated with conventional rehabilitation to enhance neuroplasticity and consequently motor function and mobility. We observed, with low evidence, that robot-assisted gait rehabilitation, alone or associated with other methods (e.g., Biofeedback EMG), leads to improvement in gait function and independence, in comparison to conventional therapy alone. Moreover, we may say that this kind of rehabilitation relieves the care to acquire the same therapeutic dosage we can deliver with traditional methods using fewer human resources. External stimulation appears to be a promising technique when TMS is associated with robotic rehabilitation. Finally, the usage of Intermittent Hypoxia associated with physical exercise seems to be the fastest method to improve gait outcome measures. Nonetheless, there is still a lack of large sample trials with good homogeneity of data and long-term follow-up. Future research should take into account different combinations of treatment to further increase motor outcomes, and therefore quality of life, in patients with SCI.

## Figures and Tables

**Figure 1 brainsci-13-00703-f001:**
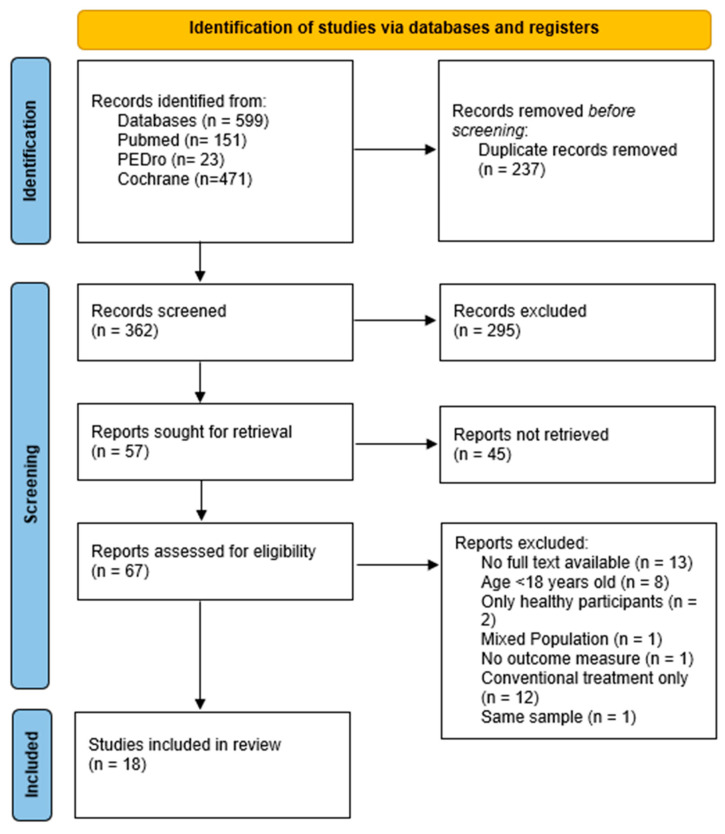
Selection process.

**Figure 2 brainsci-13-00703-f002:**
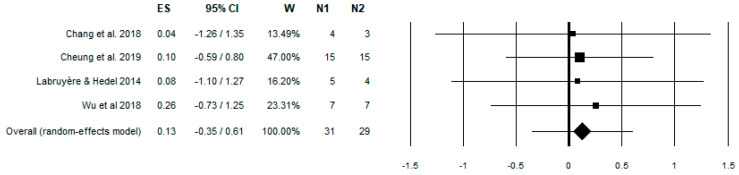
Forest plot of 10MWT at comfort speed. Effect Size (ES) 95% of confidence interval (95% CI); Weight of the effect size (W); Case Sample (N1); Control Sample (N2). Chang et al. 2018 [18], Cheung et al. 2019 [32], Labruyère & He-del 2014 [35], Wu et al. 2018 [36].

**Figure 3 brainsci-13-00703-f003:**
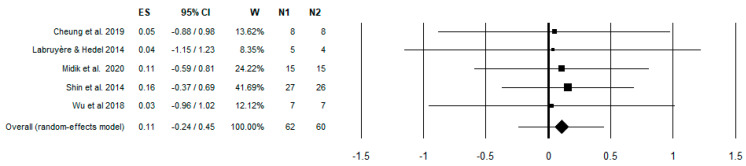
Forest plot of LEMS. Effect Size (ES) 95% of confidence interval (95% CI); Weight of the effect size (W); Case Sample (N1); Control Sample (N2). Cheung et al. 2019 [32], Labruyère & He-del 2014 [35], Midik et al. 2020 [29], Shin et al. 2014 [33], Wu et al. 2018 [36].

**Figure 4 brainsci-13-00703-f004:**
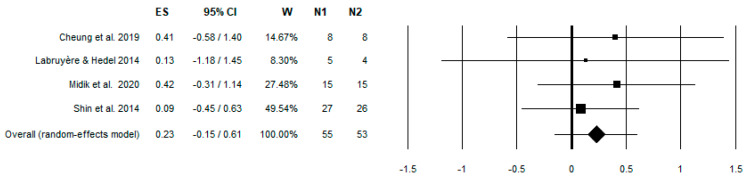
Forest plot of WISCI-II. Effect Size (ES) 95% of confidence interval (95% CI); Weight of the effect size (W); Case Sample (N1); Control Sample (N2). Cheung et al. 2019 [32], Labruyère & He-del 2014 [35], Midik et al. 2020 [29], Shin et al. 2014 [33].

**Figure 5 brainsci-13-00703-f005:**
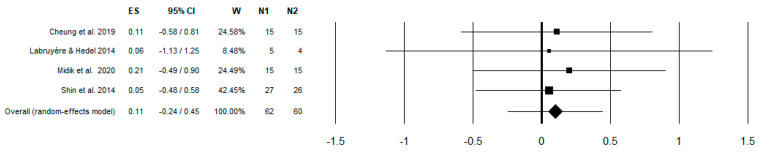
Forest plot of SCIM-3. Effect Size (ES) 95% of confidence interval (95% CI); Weight of the effect size (W); Case Sample (N1); Control Sample (N2). Cheung et al. 2019 [32], Labruyère & He-del 2014 [35], Midik et al. 2020 [29], Shin et al. 2014 [33].

**Figure 6 brainsci-13-00703-f006:**
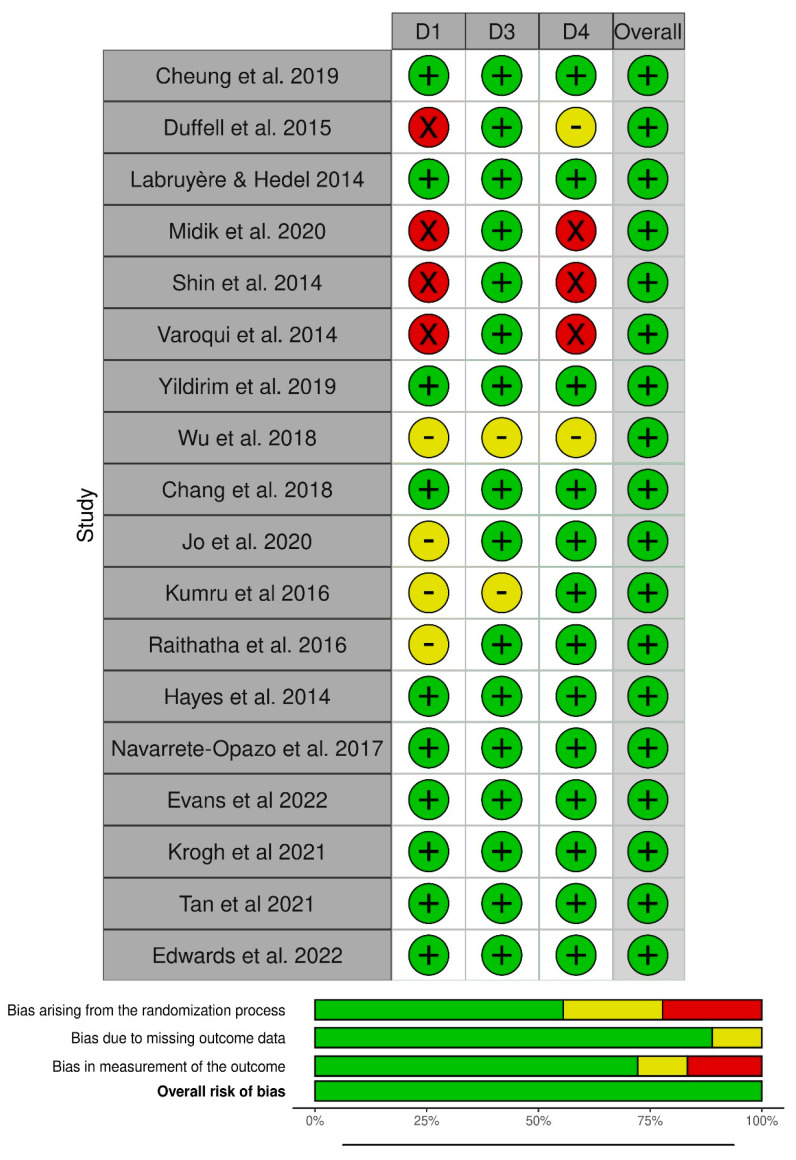
“Traffic” Light graphics of Risk of bias, RoB 2. Cheung et al. 2019 [32], Duffell et al. 2015 [34], Labruyère & Hedel 2014 [35], Midik et al. 2020 [29], Shin et al. 2014 [33], Varoqui et al. 2014 [30], Yildirim et al. 2019 [31], Wu et al. 2018 [36], Chang et al. 2018 [18], Jo et al. 2020 [42], Kumru et al. 2016 [41], Raithatha et al. 2016 [44], Hayes et al. 2014 [38], Navarrete-Opazo et al. 2017 [40], Evans et al. 2022 [45], Krogh et al. 2021 [43], Tan et al. 2022 [39], Edwards et al. 2022 [37].

**Table 1 brainsci-13-00703-t001:** PICO criteria.

Population	Adult people with stabilized spinal cord, tetraplegic, or paraplegic injuries with gait impairment.
Intervention	Rehabilitation with innovative methods
Comparison	New rehabilitation methods vs. conventional methods
Outcome	Improvement in gait parameters

**Table 2 brainsci-13-00703-t002:** RCTs selected for the systematic review.

Study	Intervention	Intervention Time	Case/Control	Outcome Measures	Mean Age ± SD Case/Control	Lesion Level	AIS	Time since Lesion	Results/Conclusions
Cheung et al. 2019 [32]	Lokomat + EMG feedback	8 weeks	15/15	10MWT 6MWT TUG WISCI-II MAS	55.6 ± 4.98/53.0 ± 12.94	<T10	C, D	<1 year	Use of EMG-biofeedback RAGT enhanced the walking performance for SCI subjects and improved cardiopulmonary function
Duffell et al. 2015 [34]	Lokomat	4 weeks	27/29	10MWT 6MWT TUG LEMS	46.6 ± 12.6/47.8 ± 13.1	<T10	C, D	>1 year	Overall, walking speed and endurance improved, with no difference between interventions. Improvements in function were achieved in a limited number of people with SCI
Labruyère & Hedel 2014 [35]	Lokomat	4 weeks	5/4	10 MWT LEMS WISCI-II FET SCIM BERG PCI	59 ± 11 *	C4-T11	C, D	>1 year	No significant differences in changes in scores between the 2 interventions, except for maximal walking speed (10MWT), which improved significantly more after strength training than after RAGT
Midik et al. 2020 [29]	Lokomat	5 days	15/15	LEMS WISCI-II SCIM-III	35.4 ± 12.1/37.9 ± 10.0	T12-L3	C, D	>3 months	Conventional rehabilitation is useful in terms of the improvement in the lower extremity motor function, walking, and functional status in men with incomplete SCI. RAGT provides greater improvement in the lower extremity motor function and functional independence.
Shin et al. 2014 [33]	Lokomat	4 weeks	27/26	LEMS AMI SCIM-III WISCI-II	43.15 ± 14.37/48.15 ± 11.49	C1-L4	D	<6 months	RAGT combined with conventional physiotherapy could yield more improvement in ambulatory function than conventional therapy alone
Varoqui et al. 2014 [30]	Lokomat	4 weeks	15/15	Distance walked in 2 min 10MWT 6MWT TUG	50.80 ± 2.12/44.65 ± 2.66	<T10	C, D	<1 year	The improvements in the kinematic and kinetic parameters of the ankle voluntary movement, and their correlation with the functional assessments, support the therapeutic effect of robotic-assisted locomotor training on motor impairment in chronic iSCI
Yildirim et al. 2019 [31]	Lokomat	8 weeks	44/44	WISCI-II FIM	32 ± 23/36.5 ± 24	C1-L4	A, B, C, D	<6 months	Robotic-assisted gait training combined with conventional therapy was found to be superior to the conventional therapy in terms of gait function and level of disability
Wu et al. 2018 [36]	3DCaLT	6 weeks	7/7	6MWT speed MAS BERG SF-36	48.4 ± 13.5/48.1 ± 4.9	C2-T10	C, D	>1 year	A greater improvement in 6-min walking distance was observed after robotic training than that after treadmill-only training
Chang et al. 2018 [18]	Ekso	3 weeks	4/3	10MWT 6WT TUG LEMS spacial-temporal parameters	56 ± 17/60 ± 2	<T12	C, D	<6 months	Improvement was observed in the 6MWT for the exoskeleton (EGT) group. Both the EGT and the conventional groups showed significant increases in right step length. The EGT group also showed improvement in stride length.
Jo et al. 2020 [42]	TMS + exercise	3 weeks	13/12 + 13 **	10MWT GRASSP MEP MVC	44.2 ± 14.8 *	C2-L3	A, C, D	>1 year	Stimulation contributed to preserving exercise gains. Our findings indicate that targeted non-invasive stimulation of spinal synapses might represent an effective strategy to facilitate exercise-mediated recovery
Kumru et al. 2016 [41]	TMS + Lokomat	8 weeks	17/17	10MWT WISCI-II LEMS	46.4 ± 15.5/48.7 ± 16.5	<T12	C, D	<6 months	A total of 20 sessions of daily high-frequency TMS combined with Lokomat gait training can lead to clinical improvement in gait in motor-incomplete SCI
Raithatha et al. 2016 [44]	tDCS + Lokomat	3 weeks	9/6	10MWT 6MWT TUG BERG SCIM-III MMT	47.5 ± 13.2	C4-L1	B, C, D	>1 year	Pairing tDCS with Lokomat can improve lower extremity motor function more than Lokomat alone
Hayes et al. 2014 [38]	IH	5 weeks	4/6 ***	10MWT 6MWT	43 ± 4 *	<T12	C, D	>1 year	IH ± walking improved walking speed and distance in patients with chronic iSCI. The impact of IH is enhanced by combination with walking, demonstrating that combinatorial therapies may promote greater functional benefits in patients with iSCI
Navarrete-Opazo et al. 2017 [40]	IH + BWSTT	4 weeks	17/16	10MWT 6MWT TUG	41 ± 17/42 ± 17	>C5	C, C	>6 months	Moderate IH (daily IH) combined with locomotor training improved walking speed and endurance in subjects with iSCI
Edwards et al. 2022 [37]	Ekso	12 weeks	9/10/6	10MWT, 6MWT, TUG, WISCI-II, NASA-Task Load Index	41 ± 10/50 ± 15	C3-L1	C, D	>1 year	Chronic SCI participants with independent stepping ability at baseline can improve clinical ambulatory status
Tan et al. 2021 [39]	IH + WALK	4 weeks	5/5 ***	10MWT, 6MWT, kinematics parameters	46 ± 18	C4-T9	C, D	>1 years	Daily AIH combined with walking practice (AIH + WALK) improved overground walking performance and intralimb motor coordination in patients with chronic iSCI
Krogh et al. 2021 [43]	TMS	4weeks	10/9	LEMS, 10MWT, 6MWT)	57 ± 8/52 ± 12	C2-L1	C, D	>3 months	High-frequency TMS may increase long-term-training-induced recovery of lower limb muscle strength following SCI.
Evans et al. 2022 [45]	tDCS + Motor Skill Training	3 days	14/11	10MWT, kinematics parameters, BBS	50 ± 10/46 ± 15	CD-T8	D	>1 year	High-frequency TMS may increase long-term-training-induced recovery of lower limb muscle strength following SCI.

* Only cumulative data available; ** Studies with double control (placebo + exercise − TCMS + exercise − TCMS); *** Crossover Trial with two blocks (IH/placebo − IH/placebo + ground training). AIS: A= Complete, B= Sensory incomplete, C= Motor incomplete, D= Motor incomplete.

**Table 3 brainsci-13-00703-t003:** Outcome measures of studies of the first group expressed as Mean ± SD.

Study	10 MWT Self-Selected Speed	10 MWT Fast	6MWT	TUG	WISCI-II	SCIM-3	LEMS
Cheung et al. 2019 [32]	0.44 ± 0.24/0.45 ± 0.24 0.44 ± 0.29/0.48 ± 0.34				14.6 ± 4.27/16.3 ± 4.95 17.0 ± 2.78/17.1 ± 2.59	73.3 ± 19.73/71.0 ± 26.32 80.0 ± 17.44/80.3 ± 17.69	35.5 ± 4.50/36.5 ± 6.16 39.4 ± 9.07/40 ± 8.89
Duffell et al. 2015 [34]		No comparable	No comparable	No comparable			35.0 ± 13.9/34.6 ± 12.3 42.6 ± 4.6/41.9 ± 5.3
Labruyère & Hedel 2014 [35]	0.62 ± 0.23/0.66 ± 0.29 0.58 ± 0.19/0.64 ± 0.23	0.79 ± 0.31/0.80 ± 0.35 0.66 ± 022/0.80 ± 0.28			14.1 ± 2.5/14.9 ± 3.1 14.4 ± 2.6/14.8 ± 2.9	88.4 ± 7.9/89.2 ± 7.6 87.9 ± 8.1/89.2 ± 7.9	40.9 ± 7.5/41.6 ± 7.3 40.4 ± 6.6/41.4 ± 6.9
Midik et al. 2020 [29]					9.8 ± 5.42 /13.7 ± 4.26 11 ± 4.26/13.6 ± 3.87	69.1 ± 18.98/79.1 ± 17.81 69.2 ± 11.62/76.2 ± 9.29	27.1 ± 12.78/28.9 ± 13.94 23.8 ± 8.91/24.4 ± 8.52
Shin et al. 2014 [33]					5.67 ± 10.97/10 ± 14.87 * 6.67 ± 12.55/9.67 ± 15.68 *	5 ± 8.61/12 ± 20.35 * 8 ± 14.12/14 ± 25.88 *	27.67 ± 21.91/35.33 ± 22.7 * 31 ± 15.69/35 ± 21.96 *
Varoqui et al. 2014 [30]	0.56 ± 0.09/0.64 ± 0.10 No available data		206.96 ± 29.57/208.87 ± 28.36 No available data	34.15 ± 9.61/27.83 ± 7.32 No available data			
Yildirim et al. 2019 [31]					5 (9)/9 (7) ** 5 (6.7)/6.5 (5) **		
Wu et al. 2018 [36]	0.33 ± 0.15/0.39 ± 0.20 0.56 ± 0.24/0.56 ± 0.24	0.48 ± 0.22/0.54 ± 0.29 0.80 ± 0.34/0.79 ± 0.35	120 ± 37/157 ± 59 control was 218 ± 92 m and 225 ± 96				
Chang et al. 2018 [18]	0.17 ± 0.01/0.22 ± 0.03 0.51 ± 0.0.28/0.55 ± 0.31		50 ± 23/67 ± 25 147 ± 87/154 ± 94	71 ± 23/55 ± 8 37 ± 17/36 ± 17			
Edwards et al. 2022 [37]	0.18 ± 0.23/0.07 ± 0.11/0.03 ±0.03	0.20 ±0.24/ 0.14 ±0.18/0.03 ± 0.13	No comparable	No comparable	No comparable		

* Data converted from median and interquartile range to mean ± SD according to Wan et al.’s 2014 formula; ** Data expressed as Median (interquartile range).

**Table 4 brainsci-13-00703-t004:** Outcome measures of studies of the second and third groups expressed as Mean ± SD.

Study	10 MWT Self-Selected Speed	10 MWT Fast	6MWT	TUG	WISCI-II	SCIM-3	LEMS	MMT
Jo et al. 2020 [42]		12.4% * 16.5% */24.5% *						
Kumru et al. 2016 [41]	2 of 15/6 of 15 ^#^ 2 of 16/4 of 16 ^#^				Comparable between the two groups		+8.2 +3.4	
Raithatha et al. 2016 [44]	0.18 ± 0.15/+0.04 ± 0.07 ^##^ 0.16 ± 0.07/+0.14 ± 0.07 ^##^		188 ± 212/+21.8 ± 51.4 4 ^##^ 184 ± 88/+132.5 ± 64.35 ^##^	38.7 ± 12.9/+ 0.6 ± 10.53 ^##^ 77.5 ± 7.0/–18.5 ± 1.98 ^##^		59.7 ± 19.5/1.2 ± 1.47 ^##^ 44.2 ± 26.5/2.7 ± 1.13 ^##^		L 4.3 ± 2.1 ** R 9.1 ± 3.8 **
Hayes et al. 2014 [38]		Not significative	+269 m +173 m					
Navarrete-Opazo et al. 2017 [40]	−20.3 ± 6.9 s −15.5 ± 4.8 s		+70.5 ± 13.2 m + 43.1 ± 10.7 m	−23.7 ± 11.1 s −22.8 ±11.5 s				
Evans et al. 2022 [45]		0.69 ± 0.51/0.83 ± 0.51						
Krogh et al. 2021 [43]		18.5 ± 30.5 /2.5 ±2.1	77.7 ± 65.5/75.6 ± 56.9	4.3 ± 3.0/3.7 ± 3.8				
Tan et al. 2022 [39]	No comparable	No comparable	No comparable					

* This study shows the necessary time to complete the 10 meters path in percentual reduction before and after the intervention with two control groups (TCMS + exercise/ placebo + exercise / TCMS). ** Within-group changes from baseline to postintervention tDCS/Sham. ^#^ Capable of completing the test. ^##^ Mean difference before and after.

**Table 5 brainsci-13-00703-t005:** Robotic treatment dosage.

Study	Weeks of Treatment	Session * Week	Sessions	Minutes	Total Minutes
Midik et al. [29]	8	3	24	30	720
Cheung et al. [32]	5	3	15	30	450
Labruyère & Hedel [35]	4	4	16	45	720
Shin et al. [33]	4	3	12	40	480
Edwards et al. [37]	12	3	36	45	1620

## Data Availability

Not applicable.

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
