# Peer review of "Gait Recovery in Spinal Cord Injury: A Systematic Review with Metanalysis Involving New Rehabilitative Technologies"

_brainsci, 2023, doi:10.3390/brainsci13050703_

Round 1

Reviewer 1 Report

Dear Authors,

thank you very much for sending the article titled: Gait recovery in spinal cord injury: a systematic review with metanalysis involving new rehabilitative technologies for the review process. The article seems interesting, but the authors should correct it according below suggestions:

- please read the article carefully and correct typos, for example, line 49 ...reorganization [8].; line 51 ...cord [10].

- I suggest writing a few sentences about the exoskeletons, which are used recently for gait recovery. The authors wrote about it in lines 116-117. I think it will be better to write about it in the Introduction section.

- In my opinion it will be great arite a few sentences about gait recovery, why it is so much important. In recent years, new models of exoskeletons have been developed. For example, in the article titled: A kinematic model of a humanoid lower limb  exoskeleton with pneumatic actuators, Acta of Bioengineering and Biomechanics, 2022, authors proposed a new models which can be used for gait recovery. Solutions of this type will appear more and more often and in my opinion it would be worth writing a few of them in Discussion section

- Lack of future research

Author Response

Thank you for the carefully review of our article.

Exoskeleton method is now cited in the introduction; however we thought going deeper in the introduction wasn’t equal regarding the other methodologies that were only cited.

We appreciated the suggestion of the article: “A kinematic model of a humanoid lower limb exoskeleton with pneumatic actuators” and we added at the end of the discussion on RAGT. As for future research, you can see our thoughts in the last lines of the article.

The article was revised by a native English speaker.

Reviewer 2 Report

Thank you for inviting me to review the paper by La Rosa et al. It is a hybrid review with meta-analysis on the use of robotics in the management of patients with spinal cord injury, and a narrative one on the use of external stimulation and intermittent hypoxia.

My main concerns about the current review are the lack of clarity in the presentation of its results and the lack of insight in the discussion section. I suggest the authors to work more on both sections, improving Table 2 and providing a new one for the narrative part of this review. In addition, the discussion should reflect the sense of the evidence found and not each article separately. This is especially relevant since the paper is introduced as a narrative review for external stimulation and intermittent hypoxia in the management of patients with spinal injury, and the current work does not sumarize the evidence in this regard. Another option would be to separate both reviews and go more in depth on each of the proposed one.

Why were previous systematic reviews excluded? There are some interesting ones by Sinovas-Alonso 2021 on gait rehabilitation in this pathology, and another by Gandara-Sambade 2017 on robotic systems for gait rehab in this population. What is new in the current systematic review? I belive the authors should emphasize the novelty of their work, as their conclusions are very similar to these two recent reviews.

A table with the selected studies on external stimulation and intermittent hypoxia is missing. In addition, this table should summarize the variables studied in each study and the main findings.

Minor comments:
Please check all quotes as there are two blanks, or none in some of them. Also, the citation number should go before the period, not after.

Line 53: Please add "and" before "robot asssited gait training".

Table 2. I believe the footnotes have been moved up to the title. Please check. In addition, this table would look better on a horizontal sheet.

Line 179: This sentence should not be separated from the previous paragraph.

Line 185: please remove the period.

Line 212: a word is missing after "motor"

Author Response

Thank you for the helpful comments,

we have improved the clarity of presentation of results and also discussion (by adding a comparison with the existing review literature), although we have maintained (and better explained) the two modalities of review. Indeed, the search was systematic, but metanalysis done only on robotics because of the lack of sufficient data for the other methodologies.

We haven’t utilized other systematic reviews data because we preferred to use raw data and then analyze them by ourselves.

Unfortunately, we observed that Sinovas-Alonso is focused on outcome measures and not rehabilitative methods; the second one is written in Spanish, as you can see, we haven’t analyzed Spanish written papers.

We enhanced discussion and results with comments about other reviews.

We made two different tables for metanalysis (table3) and narrative (table 5). Also table 2 was improved. We corrected al citation style typos; regarding the imagination of the tables, we sent them in horizontal, then was transformed by editorial office in vertical. We agree to horizontal format.

Minor points have also been addressed

Round 2

Reviewer 2 Report

I would like to thank the authors for their efforts to improve the wording and clarity of the discussion and the introduction, as well as the inclusion of new tables and the improvement of the existing ones.
I am now satisfied with the changes made.